# The Controversial Roles of ADP-Ribosyl Hydrolases MACROD1, MACROD2 and TARG1 in Carcinogenesis

**DOI:** 10.3390/cancers12030604

**Published:** 2020-03-05

**Authors:** Karla L.H. Feijs, Christopher D.O. Cooper, Roko Žaja

**Affiliations:** 1Institute of Biochemistry and Molecular Biology, RWTH Aachen University, Pauwelsstrasse 30, 52074 Aachen, Germany; rzaja@ukaachen.de; 2Department of Biological and Geographical Sciences, School of Applied Sciences, University of Huddersfield, Queensgate, Huddersfield West Yorkshire HD3 4AP, UK; c.d.cooper@hud.ac.uk

**Keywords:** ADP-ribosylation, cancer, macrodomain, ADP-ribosyl hydrolase, PARP, ARTD, MACROD1, MACROD2, TARG1

## Abstract

Post-translational modifications (PTM) of proteins are crucial for fine-tuning a cell’s response to both intracellular and extracellular cues. ADP-ribosylation is a PTM, which occurs in two flavours: modification of a target with multiple ADP-ribose moieties (poly(ADP-ribosyl)ation or PARylation) or with only one unit (MARylation), which are added by the different enzymes of the PARP family (also known as the ARTD family). PARylation has been relatively well-studied, particularly in the DNA damage response. This has resulted in the development of PARP inhibitors such as olaparib, which are increasingly employed in cancer chemotherapeutic approaches. Despite the fact that the majority of PARP enzymes catalyse MARylation, MARylation is not as well understood as PARylation. MARylation is a dynamic process: the enzymes reversing intracellular MARylation of acidic amino acids (MACROD1, MACROD2, and TARG1) were discovered in 2013. Since then, however, little information has been published about their physiological function. MACROD1, MACROD2, and TARG1 have a ‘macrodomain’ harbouring the catalytic site, but no other domains have been identified. Despite the lack of information regarding their cellular roles, there are a number of studies linking them to cancer. However, some of these publications oppose each other, some rely on poorly-characterised antibodies, or on aberrant localisation of overexpressed rather than native protein. In this review, we critically assess the available literature on a role for the hydrolases in cancer and find that, currently, there is limited evidence for a role for MACROD1, MACROD2, or TARG1 in tumorigenesis.

## 1. ADP-Ribosylation Reactions

Post-translational modifications of proteins can have a myriad of consequences: changing interactomes, stability, localisation, or activity are just a few examples. To date, more than 200 types of modification are known [1]. The addition of ADP-ribose moieties to a protein was first reported in the 1960s when chains of (poly)ADP-ribose (PAR) were identified on target proteins, termed poly(ADP-ribosyl)ation (PARylation) [2]. Since then, PARylation has been intensively studied, leading to the identification of crucial roles in the repair of DNA damage and to the development of specific inhibitors that are utilised in the clinic to treat diverse cancers [3]. In contrast, much less is known about its smaller sibling, mono(ADP-ribosyl)ation (MARylation), where only single ADP-ribose units (MAR) are conjugated into proteins. In 2008, it was first noted that PARP10/ARTD10 has MARylation rather than PARylation activity [4] and, lately, it has become clear that only the minority of PARP enzymes are capable of PARylation [5]. Moreover, not only protein, but also DNA has been identified as a target for ADP-ribosylation catalysed by PARP3 [6,7]. Following PARP1 and PARP2, PARP3 has also been described as a potential therapeutic target in certain cancers [8,9,10,11]. The most recent addition to the spectrum of ADP-ribosylated molecules is RNA, which can be modified in vitro by multiple PARPs [12].

One of the best-studied but still barely understood MARylating enzymes is PARP10. PARP10 was initially identified as an interactor of MYC [13] and, later, of ubiquitinated proliferating cell nuclear antigen (PCNA) [14,15]. Its overexpression leads to cell death in HeLa cells through an unknown mechanism [16]. In contrast, in non-transformed RPE-1 cells, overexpression leads to stimulation of cell growth [17]. It has been given a role in replication and DNA damage [15,17,18] as well as mitochondrial metabolism [19], but it is not clear what the relevant substrates of this enzyme are. Potential substrates were identified using protein microarrays [20], which was performed for the PARylating PARP2 [21]. However, partially due to a lack of antibodies at that time, these modifications were not verified to occur in cells. PARP10 was also reported to be a regulator of NF-kB signaling [22]. PARP14 functions as a co-activator of STAT6 and has been described as an anti-cancer and anti-inflammatory target, reviewed in more detail elsewhere [23,24]. PARP16 is located at the endoplasmic reticulum (ER) and plays a role in the unfolded protein response [25,26]. The most recent function identified for MARylation is its role in the anti-viral defence. PARP7, PARP10, and PARP12 gene expression is induced by interferon-α and ADP-ribosylation mediated by these PARPs may function to counteract viral proliferation [27,28]. In addition to enzymes of the PARP family, several sirtuins (SIRTs) appear to be able to ADP-ribosylate substrates in addition to their better-studied NAD^+^-dependent deacylase activity [29,30]. The first research showing that SIRTs possess MARylation activity described it as a very weak activity [31]. Other papers, focusing on single enzymes, showed that the mitochondrial SIRT4 MARylates glutamate dehydrogenase to downregulate insulin secretion [32]. The nuclear SIRT6 modifies PARP1 [33] and BAF10 [34] to regulate their activities and, lastly, the nucleolar SIRT7 was reported to MARylate histones p53 and ELK4 [35]. To date, very little information is available about the MARylating activity of the SIRTs. The functional consequences of MARylation in normal cell physiology, thus, appear to be very diverse, reviewed extensively elsewhere [36]. Herein, we focus on the enzymes reversing the intracellular MARylation reaction: the macrodomain-containing hydrolases.

## 2. Macrodomains

Intimately linked to ADP-ribosylation is a protein structural module known as the macrodomain, named for the unusually large histone macroH2A from which it was identified [37]. The macrodomain is a protein fold formed around a central β-sheet, flanked by α-helices that forms a pocket where ADP-ribose can bind [38,39]. These pockets are slightly different between the macrodomain-containing proteins with the consequence that the macrodomain-containing protein family can be subdivided. Some macrodomains bind to PAR and others bind to MAR [40,41,42,43].

Macrodomains not only serve as binding modules to mediate protein interactions depending on ADP-ribosylation, but a number of them have catalytic activity. Poly(ADP-ribosyl)glycohydrolase (PARG) is able to degrade PAR chains by cleaving the glycosidic bond between adjacent ADP-ribose moieties. However, it cannot remove the final ADP-ribose attached to the protein [44]. Three other proteins, MACROD1, MACROD2, and TARG1, are not active toward PAR but are capable of removing the ADP-ribose moiety connected to the protein partner, presumably by cleaving an ester bond between ADP-ribose and an acidic receptor amino acid [45,46,47]. They can, therefore, both reverse any MARylation as generated by PARP enzymes, and also remove the last ADP-ribose left behind by PARG. Of the 12 human macrodomain-containing proteins known today, 11 proteins were identified by sequence homology [39,43]. The macrodomain of PARG was only recognised after the structure had been solved [44], which indicates the possibility that more macrodomain-containing proteins remain to be identified. A recent bioinformatics approach has suggested that the largely uncharacterised protein C12orf4 harbours a macrodomain, or at least a macro-like domain, but this remains to be confirmed experimentally [48].

The macrodomain is an evolutionary well-conserved module [39]. For instance, recent work has shown that certain alpha-viruses encode a protein with a macrodomain-like fold [49], which displays catalytic activity towards MARylated proteins [50,51,52]. The existence of such proteins enhances the evidence connecting existing links between MARylation and immunity [53]. In this review, we will focus specifically on potential oncogenic functions of the mono(ADP-ribosyl)hydrolases MACROD1, MACROD2, and TARG1. It should also be noted, however, that other macrodomain-containing proteins such as PARG and CHD1L (ALC1) have been previously linked closely to cancer [54,55]. 

## 3. Macrodomain-Containing Hydrolases: MACROD1, MACROD2, and TARG1

MACROD1, MACROD2, and TARG1 are relatively small proteins (Table 1) without any notable other domains beside the macrodomain. The first catalytic activity of MACROD1, MACROD2, and TARG1 that was recognised was their deacetylation of *O-*acetyl-ADP-ribose (*O*AADPR) to generate acetyl and ADP-ribose [56,57]. *O*AADPR arises as a by-product of SIRT-mediated deacetylation in which the acetyl group is transferred onto ADP-ribose while releasing nicotinamide, and may have important biological functions [58]. Two years later, three labs reported the activity of MACROD1, MACROD2, and TARG1 in removing ADP-ribose from protein substrates, which was proposed to be limited to removal of ADP-ribose from acidic amino acids [43,45,46,47]. Serine and arginine MARylation were suggested to be removed by ARH3 and ARH1, respectively [59,60,61,62], which are structurally different proteins that do not contain a macrodomain. In 2017, MACROD1, MACROD2, and TARG1 were reported as being capable of removing ADP-ribose from both modified single-stranded DNA [63] and, in 2018, also being capable of removing ADP-ribose from MARylated single-stranded RNA [12]. Lastly, in 2019, the hydrolysis of α-NAD was added to the list of their in vitro activities [64].

To date, it is not known which of these activities is relevant for their functionality in cells, either in normal physiology nor in pathological conditions. This is predominantly due to the fact that it is not well known how prevalent their in vitro substrates are in cells or what their cellular functions may be. This is partially due to the fact that antibodies or modules recognising MARylation have been developed only recently [65,66,67]. Before these technical developments, it was not possible to confirm with antibodies that MARylation of proteins takes place in cells, and, hence, reversal mediated by these macrodomain-containing proteins could also not be assessed. Currently, it is still not straightforward to estimate endogenous MARylation levels in cells, as no inhibitors were available for MACROD1, MACROD2, or TARG1. During experimental procedures such as cell fixation or lysis, MARylation could possibly be removed by these enzymes and, therefore, go undetected. One study was performed to identify MACROD1 inhibitors using an AlphaScreen assay, where inhibitors in the micromolar range were identified. These can serve as lead compounds for further optimisation, but are likely not specific enough yet to be used for studies in cells due to their other activities at this concentration [68]. Another study identified an allosteric inhibitor of macrodomain 2 of PARP14, which blocks its binding to ADP-ribose. However, it was not tested whether the identified compound might inhibit one of the macrodomains with catalytic activity [69].

The roles of MACROD1, MACROD2, and TARG1 in normal cell biological processes are not well understood. They appear to be expressed and localised in different tissues and intracellular locations, which clearly gives them unique roles to play despite their biochemical similarity. This also explains why loss of TARG1 alone can lead to a neurodegenerative phenotype [45]. MACROD1 resides in mitochondria [40,70,71], but, depending on the tag used for overexpression, will reside exclusively in the nucleus instead [70]. MACROD2 displays a more diffuse cytoplasmic/nuclear distribution [70,75] and overexpressed TARG1 appears to be present in nucleoplasm, nucleoli, and cytoplasmic stress granules [14,70]. TARG1 and MACROD1 are expressed throughout the different tissues with an enrichment of MACROD1 in skeletal muscle at both the protein and RNA level [70]. MACROD2 has, thus far, only been detected in human neuroblastoma cells and in mouse cortical neurons [70,76]. Several reports have correlated polymorphisms or deletions within the *MACROD2* gene with autism-spectrum disorders [77,78,79,80]. It is not clear, however, whether the *MACROD2* gene product itself or surrounding genes are responsible for this association even though elevated protein expression in neurons supports a potential brain-specific function of MACROD2 [76]. However, a long non-coding RNA has been identified within an intron of the *MACROD2* gene, which is more highly expressed in most tissues investigated than the *MACROD2* mRNA, and, thereby, potentially confuses correlations of mutations in the *MACROD2* gene and phenotypes [81]. In another genome-wide association study, *MACROD2* was identified as a factor influencing vascular-adhesion protein-1 (VAP-1) levels, which the authors confirmed by knockdown of MACROD2. This leads to lower VAP-1 expression in adipocytes, presumably through the transcriptional regulation of VAP-1 [82]. MACROD2 was also reported to leave the nucleus upon DNA damage, dependent on phosphorylation by ATM [75], whereas TARG1 localises from nucleoli to nuclear sites of damage [14], which both have an unclear functional relevance. More data may be found on MACROD1, which is also known as leukaemia-related protein 16 or LRP16. MACROD1 has been attributed to a number of functions in the nucleus, such as co-activation of the androgen receptor [83], counteracting PARP7-mediated MARylation in the nucleus [84,85], and activation of NF-kB signalling [86,87,88]. MACROD1 was also reported as an enhancer of oestrogen receptor signalling [89], and is upregulated after stimulation of cells with oestrogen [72,74,90]. No regulatory factors have been identified yet for MACROD2 and TARG1. A BioID interaction screen, which identifies proteins in close proximity to the protein of interest [91], identified many proteins involved in nuclear/cytoplasmic and mitochondrial nucleic acid metabolism as interactors of TARG1 and MACROD1, respectively [70]. Whether these proteins are MARylated and serve as a substrate of TARG1/MACROD1 remains to be determined. It is possible that, despite spatio-temporal restrictions, MACROD1 and TARG1 in their respective compartments are involved in similar signalling networks, converging on the regulation of cellular nucleic acids. The physiological functions of the three enzymes remains elusive. Most puzzling perhaps is that, at the moment, despite the mitochondrial localisation of MACROD1, the majority of reports describe nuclear functions. Future work will need to address this apparent discrepancy.

## 4. Mono(ADP-ribosyl)ation in Cancer

The post-translational modification poly(ADP-ribosyl)ation has been intimately linked to cancer before [92,93], as have other post-translational modifications such as phosphorylation [94]. In BRCA1/BRCA2-deficient patients, PARP1 inhibitors have been shown effective specifically against the tumour cells applied in the clinic. However, this is one of the rare examples of a synthetic lethal interaction [95]. Better understanding of the processes regulated by MARylation will provide opportunities for further drug development, as is exemplified by current research into the potential of PARP14 as a drug target [24]. Little is known about the potential role of the mono(ADP-ribosyl)hydrolases in cancer. *MACROD1*, *MACROD2*, and *OARD1* (TARG1) exhibit mutations only in 0.9%, 2.6%, and 1% of cancer patient samples, respectively, from over 1000 samples in the cBioPortal curated dataset [96]. The fact that *MACROD2* mutation rates are twice the rates seen for *MACROD1* and *OARD1* likely reflects that the *MACROD2* gene is larger (Figure 1), and is located at a known fragile region [97]. A specific recurring deletion of exon 6 has been observed in esophageal squamous cell carcinoma and gastric cancer [98]. For *MACROD1*, a *RUNX-MACROD1* fusion was identified in leukaemia [99] and, for *MACROD2*, a *PDGFRA-MACROD2* fusion in a pleomorphic sarcoma [100]. Despite the low number of identified mutations in patient tumour samples, several reports have correlated *MACROD1* or *MACROD2* expression levels with the clinical outcome, as described in the next paragraphs.

The gene structure of *MACROD1*, *MACROD2*, and *OARD1* is shown schematically. More lncRNAs are present. However, only RPS10P2-201 is displayed since it has been shown to be relevant RPS10P2-201 [81]. CDS = coding sequence. UTR = untranslated region.

## 5. MACROD1 in Cancer

A number of studies have addressed a potential oncogenic function of MACROD1 and started deciphering the molecular mechanism underlying observed effects. MACROD1 expression was shown to be upregulated by oestrogen [73,74], which leads to several studies of the role of MACROD1 in tumours with a differential oestrogen status. MACROD1 overexpression in the oestrogen and progesterone receptor positive Ishikawa cells, derived from an endometrial cancer, had no effect on cell proliferation. It did, however, enhance the invasiveness of these cells as measured by transwell assays [101]. Mechanistically, the authors propose a mechanism wherein MACROD1, dependent on oestrogen, blocks recruitment of ERα to the E-cadherin promoter, which lowers E-cadherin expression and, through this, enhances invasiveness. shRNA-mediated knockdown of MACROD1 achieved the opposite effect by enhancing E-cadherin expression [101]. This implies that MACROD1 can be an important factor in metastasis. These findings appear to contradict an earlier report where overexpression of MACROD1 in MCF7 cells, which are oestrogen-responsive breast cancer cells, showed an effect on cell growth. It led to enhanced proliferation [74]. A later report studying MACROD1 in 293T cells showed that knockdown of MACROD1 sensitized cells to TNFα-induced apoptosis [86]. From these studies, it is, thus, not clear what effect loss or gain of MACROD1 has on cell physiology. Recent work has demonstrated that fusing an N-terminal tag, such as GFP, to MACROD1 leads to a nuclear localisation, whereas C-terminally labelled and an endogenous protein appear to be exclusively localised in mitochondria [70]. Mass spectrometry datasets have also detected MACROD1 in the mitochondria [102]. This does not exclude the possibility that, under specific circumstances, MACROD1 may re-localise to the nucleus. In pathogenic conditions, such as the presence of a *RUNX-MACROD1* fusion protein that was identified in leukaemia [99], the protein likely also localises to the nucleus instead of the mitochondria, as the RUNX fusion will mask the mitochondrial targeting sequence, comparable to the localisation after labelling with an N-terminal GFP tag. Unfortunately, some of the studies investigating MACROD1 have either used N-terminally tagged fusion constructs [84,85,88], or have not stated clearly how the fusion proteins were generated [103]. Furthermore, the majority of applied antibodies show multiple bands in the western blot and, hence, are not suitable for immunohistochemistry (IHC) or immunofluorescence (IF) (Table 2). It will be worthwhile to repeat some of the studies of MACROD1′s molecular function in carcinogenesis to clarify whether unlabeled MACROD1 overexpression leads to enhanced cell growth, or whether this effect depends on the tumour background and also to study knockdown/knockout systems more thoroughly.

Despite these technical challenges, multiple studies have correlated tumour growth with MACROD1 expression in either xenograft models or in patient cohorts. Overexpression of MACROD1 in the hepatocellular carcinoma lines HepG2 and MHCC-97L leads to a decrease in cell growth and metastatic potential, as measured by transwell assays [103]. Furthermore, when delivered into nude mice, cells overexpressing MACROD1 lead to a decreased tumour volume and have a lower metastatic potential compared to cells without overexpressed MACROD1 [103]. With the information available, it cannot be distinguished whether this is a genuine effect or an artefact due to forced nuclear localisation of the overexpression construct [103]. In a similar set of experiments in the pancreatic carcinoma cell lines, Panc1, CFPAC1, Bxpc3, SW1990, AsPC1, and HPDE6-C7, opposite results were achieved [105]. Knockdown of MACROD1 leads to enhanced apoptosis and decreased cell growth. However, only one shRNA construct was used, so any effects seen could be potentially off-target. Cells overexpressing the same MACROD1 construct as in the study described before [103] grew faster and were more resistant to apoptosis. Xenograft experiments show the same trend. Cells lacking MACROD1 had a lower tumour-forming potential and higher survival rate. Cells overexpressing MACROD1 had a higher tumour volume and a lower survival rate [105]. The authors do not comment on the opposing effects in these two tumour types, but agree that larger-scale studies are required to verify these findings. A third study shows IHC of lung tumour samples with an antibody of an unknown source and specificity [104]. High MACROD1 expression, as measured by IHC, was reported to correlate with a negative outcome in colorectal carcinoma and in gastric carcinoma. However, the antibody used was generated by the authors and not validated by a western blot [106,107]. CRISPR-mediated *MACROD1* knockout rhabdomyosarcoma cells [70] and *MACROD1* knockout mice appear viable [71], which makes it unlikely that loss of MACROD1 has a drastic growth inhibitory or developmental effect.

In conclusion, most of the data available on an oncogenic function of MACROD1 rely on poorly characterised antibodies, unclear overexpression constructs or a single shRNA construct. The majority of these studies would need to be reproduced with a more thorough characterisation and description of the materials used to be able to draw deeper conclusions. Several studies agree that MACROD1 expression can be induced by oestrogen [73,74,89,90,101], but it remains unclear what the effect of this overexpression is on cells. The *RUNX-MACROD1* fusion identified in leukaemia may provide an important hint at a potential pathologic function. It is possible that the physiological mitochondrial localisation of MACROD1 can turn into a pathogenic nuclear one, where it aberrantly acts as a transcriptional activator. It will be interesting to see whether more instances can be identified where such fusions are present. Alternatively, other masking events may occur in cells, such as binding by interaction partners or PTMs, which, thereby, redirects the protein to the nucleus for a physiological function.

## 6. MACROD2 in Cancer

A number of analyses suggest MACROD2 may play a potential role in cancer. MACROD2 copy number is increased in three different tamoxifen-resistant MCF7 breast cancer cell lines, prompting the authors to analyse MACROD2 expression in patient samples. In oestrogen receptor-positive tissues of breast cancer patients with a recorded tamoxifen-resistance, however, three patients displayed a decreased copy number, whereas the other two patients showed an increase. Using IHC with a custom antibody, varying levels of MACROD2 were detected in primary and secondary tumour tissues of these patients, collectively showing that MACROD2 may be overexpressed in cancer tissues. MCF7 and T47D cells with exogenously overexpressed MACROD2 grow faster than control cells in media containing tamoxifen, which implies that MACROD2 confers tamoxifen-resistance to the cells and appear to be stimulated by tamoxifen, as demonstrated by faster growth in medium with tamoxifen than without [110]. Conversely, tamoxifen-resistant MCF7 clones with shRNA-mediated knockdown of MACROD2 become more sensitive even though it does not completely reverse resistance [110]. This may be mediated by the activation of oestrogen-regulated genes in response to tamoxifen in cells overexpressing MACROD2. Lastly, cells stably expressing shRNA to knockdown MACROD2 grow markedly slower in nude mice [110]. The authors argue that, as a cancer specific fragile site, the MACROD2 gene can be lost, but this fragility also allows for amplification in the specific case of ER-positive breast cancers treated with tamoxifen to incur resistance. A more recent study showed that, in approximately one-third of colorectal carcinomas investigated, heterozygous or homozygous losses were mapped within the *MACROD2* gene in which the majority are intragenic microdeletions mapping to a region in exons 4 to 5 [111]. To test whether MACROD2 deficiency can promote tumourigenesis, *MACROD2* knockout was introduced into an adenomatous polyposis coli protein (Apc) mouse model. Mutations in the APC tumour suppressor are a major driver of sporadic colorectal cancers, where it could be shown that even haploinsufficiency of *MACROD2* leads to more and larger adenoma formation [111]. Furthermore, human cells transplanted into nude mice displayed increased tumour growth when lacking *MACROD2*, but a reduced tumour growth when MACROD2 was overexpressed [111]. The underlying mechanism was suggested to be impaired PARP1 activity in the *MACROD2*^-/-^ cells, which leads to increased sensitivity of DNA damage and, ultimately, causes enhanced chromosomal instability [111,112].

These findings appear paradoxical with the previous report, where loss of MACROD2 impairs cell growth from a loss of resistance to tamoxifen. *MacroD2* knockout mice do not show altered survival rates after sub-lethal irradiation compared to wildtype mice, which indicates that loss of MACROD2 alone is not sufficient to drive tumourigenesis triggered by DNA damage [109]. It is not clear whether loss or overexpression of MACROD2 contributes to tumourigenesis, or whether both can drive tumour growth dependent on conditions, such as the functionality of the DNA damage repair systems or the oestrogen receptor status. In an investigation of stage-III colon cancer, MACROD2 expression determined by immunohistochemistry was found to correlate with poor survival [113]. A human protein atlas antibody (HPA049076) was used for this work, which has been retracted by the company in the meantime and did not appear to be validated in any way, such as western blotting or siRNA-mediated knockdown. It is, thus, not clear whether this antibody recognises MACROD2 at all, or whether it recognises additional proteins, which may be upregulated in the samples analysed. Altogether, it appears that loss of MACROD2 as such is not sufficient to drive tumourigenesis, but may have an additive effect in models prone to tumour formation such as loss of APC in colorectal cells. More studies are urgently needed to clarify the exact role of MACROD2 in both the onset of cancer as well as in the response of existing tumours to therapies.

## 7. TARG1 in Cancer

The only phenotype associated with TARG1 expression is neurodegeneration, occurring due to a mutation that leads to a truncated protein with a disrupted macrodomain, and, thereby, loss of catalytic activity [45]. It is not clear whether this neurodegeneration is a result of a potentially toxic truncated or unfolded protein or from loss of hydrolase activity. Knockdown of TARG1 leads to a decrease in 293T cell proliferation and a slight increase in senescence in U2OS cells, which are derived from an osteosarcoma [45]. CRISPR-mediated knockdown of TARG1 does not influence HeLa or U2OS proliferation [14,70], which leaves it unclear in which setting TARG1 is required for cell growth. Overexpression does not lead to changes in cell proliferation [14,70]. We could not identify any data linking TARG1 to cancer, nor see elevated expression or mutation in databases such as COSMIC. Based on expression levels in databases and these experimental results, at this stage, it does appear unlikely that TARG1 is involved in cancer even though further experimentation is required.

## 8. Conclusions

Despite the presence of several publications reporting a correlation between MACROD1 or MACROD2 levels and cancer development or progression, it is not clear at the moment that they have a causative role. MACROD2 is potentially relevant in ER-positive, tamoxifen resistant breast cancers where it may confer resistance to treatment. However, larger cohorts of patient samples need to be analysed to further substantiate these initial findings. Loss of *MACROD2* in an APC null background potentially stimulates tumour formation. However, loss of *MacroD2* alone does not have any effect in a knockout mouse model. Detailed studies are required to determine how loss of MACROD2 might cooperate with loss of APC to drive cancer. MACROD1 levels are upregulated by oestrogen with unclear consequences for cells. Yet, it may down-regulate E-cadherin and, thereby, promote metastasis. If confirmed, this may be an attractive therapeutic target intending to keep the cancer dormant and prevent metastasis.

Conflicting data show that overexpression and knockdown of MACROD1 have no effect, no stimulus, and no inhibition of cell growth, which implies that, perhaps, inhibiting MACROD1 may not have detrimental effects for the whole organism but rather may be dependent on the tumour background and, thus, may represent a valid drug target. The effects observed may, however, be dependent on the constructs used as well as on the specific cell-types and have to be studied in more detail. Future work will have to dissect in which context loss or gain of MACROD1 may be driving aspects of cancer growth. The recent development and characterisation of more specific antibodies, the ongoing improvements of mass spectrometric measurements of MARylation, and also the attempts at making specific inhibitors for both transferases and hydrolases should allow a more detailed analysis of their (patho-)physiological function in the near future. The partially paradoxical findings described will undoubtedly be clarified with better validated tools to determine the extent to which MACROD1, MACROD2, and TARG1 are relevant for tumourigenesis in order to establish their potential as drug candidates.

## Figures and Tables

**Figure 1 cancers-12-00604-f001:**
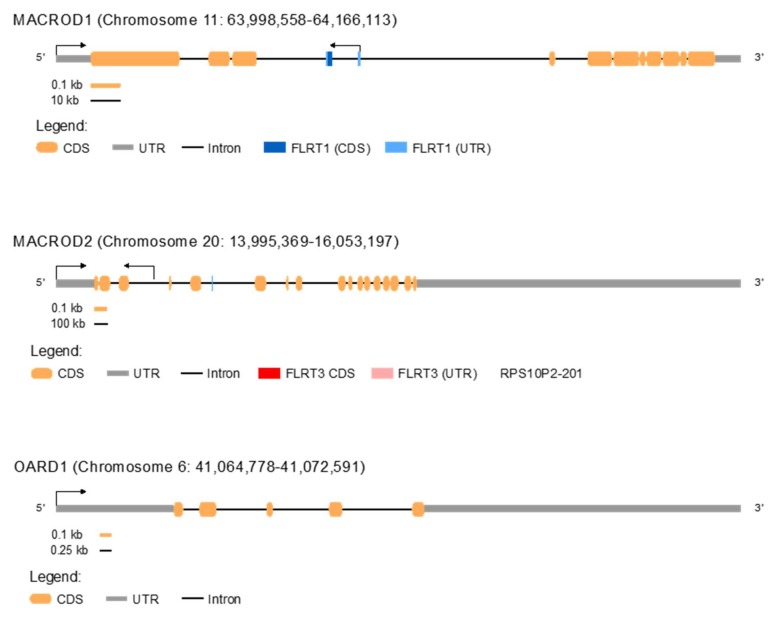
Overview of the *MACROD1*, *MACROD2*, and *OARD1* gene structure.

**Table 1 cancers-12-00604-t001:** Summary of nomenclature, expression, and localisation data of MACROD1, MACROD2, and TARG1.

Protein Name	Alternative Names	Gene Name	MW Predicted	MW Observed	Intracellular Localisation	Expression	Regulation
MACROD1	LRP16	*MACROD1*	35.5 kDa	27 kDa* [70,71]	Mitochondrial [40,70,71]	Ubiquitous expression, enriched in skeletal muscle [70,71]	mRNA expression is induced by oestrogen [72,73,74]
MACROD2	C20orf133	*MACROD2*	47 kDa	50 kDa [70]	Diffuse nuclear, cytoplasmic [70,75]	So far detected only in the brain [70,76]	Phosphorylation by ATM upon DNA damage induces translocation to cytoplasm [75]
TARG1	C6orf130	*OARD1*	17 kDa	17 kDa [14,70]	Nuclear, nucleolar, stress granular [14,45,70]	Ubiquitous expression [70]	Leaves nucleoli upon DNA damage [14]

* The predicted molecular weight of MACROD1 is 35 kDa. However, due to cleavage of the N-terminus upon translocation into mitochondria, the protein detected in western blotting (WB) is smaller [70].

**Table 2 cancers-12-00604-t002:** Summary of studies studying the effect of MACROD1 and MACROD2 protein levels on cell and/or tumour growth.

Cancer Tissue	Protein	Expression	Effect/Prognosis	Antibody Used	Reference
Neuroendocrine lung tumours	MACROD1	Elevated	Poorer survival	Monoclonal rabbit antibody against LRP16 Not further specified or validated	[104]
Hepatocellular carcinoma	MACROD1	Overexpressed: N- or C- tag not specified	Lower cell and tumour growth	Santa Cruz goat polyclonal This antibody is not available anymore. Whole blots are not shown. It was not validated with siRNA	[103]
Pancreatic carcinoma	MACROD1	Overexpressed: N- or C- tag not specified	Higher cell and tumour growth	Abcam rabbit polyclonal This antibody recognises multiple bands in WB and is thus not suitable for IHC/IF	[105]
Colorectal carcinoma	MACROD1	Elevated	Poorer survival	Polyclonal rabbit antibody generated by the authors’ institute Not further specified or validated	[106]
Gastric carcinoma	MACROD1	Elevated	Poorer survival	Polyclonal rabbit antibody generated by the authors’ institute Not further specified or validated	[107]
Breast cancer	MACROD1	MACROD1 expression quantified as either positive or negative	MACROD1 expression was higher in patients with advanced stages	LRP16 rabbit anti-human antibody, source not given Not further specified or validated	[108]
Endometrial cancer	MACROD1	Overexpressed	No effect on proliferation but enhanced invasion	Antiserum generated in rabbits against amino acids 83-324	[101]
Breast cancer	MACROD1	Overexpression	Increased proliferation	Not specified	[74]
Colorectal carcinoma	MACROD1	Non-tagged and N-terminal flag-tagged overexpression	Confers resistance to chemotherapeutics	Antibody used for IHC and WB not specified, recognises bands at ±35 and ±45 kDa	[88]
Tumours induced in mice by sublethal irradiation	MACROD2	Knockout mice	No difference between wildtype and MacroD2^-/-^	Thermofisher PA5-45950 This antibody recognises bands at ±38 kDa and ±22 kDa	[109]

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
