# Peer review of "The Controversial Roles of ADP-Ribosyl Hydrolases MACROD1, MACROD2 and TARG1 in Carcinogenesis"

_cancers, 2020, doi:10.3390/cancers12030604_

Round 1

Reviewer 1 Report

The manuscript “The controversial roles of mono (ADP-ribosyl) hydrolases MacroD1, MacroD2 and TARG1 in carcinogenesis,” by Feijs et al. is an excellent review of the current knowledge of connecting these macrodomain proteins to cancer.  Since they only focus on the mono-ADP-ribosylation binding macrodomain proteins, it allows their discussion to remain tight.  They do a good job reviewing the biology/cancer connections and I particularly liked their discussion of the potential controversy over the localization of MacroD1 – is it mitochondrial or nuclear?  Overall, this will serve as an excellent reference for those wishing to understand the field at this point, and what questions might be relevant to explore moving forward.  Before the manuscript is accepted, I would recommend a few corrections/additions to make it more suitable for publication.

First, there is a growing amount of literature showing that sirtuins such as SIRT4 (mitochondrial), SIRT6 (nuclear), and more recently SIRT7 (nuclear) display mono-ADP-ribosylation (mADPr) activity. Feijs et al. should include these proteins (and references) with their list of PARPs that show mADPr since it is highly germane to their focus – ie macrodomains that hydrolyze mADPr. A minor point- on page 3 lines 113-114, the authors mention that “no inhibitors exist for MacroD1, MacroD2, or TARG1” In fact, recently, some inhibitors for MacroD1 and MacroD2 were published 1,2, and this should be included/cited Minor typos etc: In the abstract, on page 1, lines 13-14, it’s probably more appropriate to replace “multiple ADP-ribose moieties (PARylation) ” with “poly-ADP-ribosylation (PARylation), and “only one 13 unit (MARylation)” with “mono-ADP-ribosylation (MARylation)” Please check all instances of symbols such as “beta” and “alpha” on page 2, line67 or page 3 line 105- in my downloaded version they do not appear Page 7 line 219 – “Using” should be deleted Page 3 lines 222-223, the sentence “ Cells overexpressing the same MacroD1 222 overexpression construct grew faster and were more resistant to apoptosis.” Could be re-written using only a single overexpressing term

1          Haikarainen, T., Maksimainen, M. M., Obaji, E. & Lehtio, L. Development of an Inhibitor Screening Assay for Mono-ADP-Ribosyl Hydrolyzing Macrodomains Using AlphaScreen Technology. SLAS Discov 23, 255-263, doi:10.1177/2472555217737006 (2018).

2          Schuller, M. et al. Discovery of a Selective Allosteric Inhibitor Targeting Macrodomain 2 of Polyadenosine-Diphosphate-Ribose Polymerase 14. ACS Chem Biol 12, 2866-2874, doi:10.1021/acschembio.7b00445 (2017).

Author Response

We would like to thank the reviewer for the positive comments.

The indicated references to inhibitor studies were omitted inadvertently and have been added to the revised version.

We have also added a paragraph about potential MARylation activity of the SIRTs for completeness. We had not included this in the first version, because the evidence that this activity is relevant in cells is still relatively weak.

All minor points have been addressed.

Reviewer 2 Report

The review presented by KHL Feijs and co-authors apposes the different reports on the macro-containing ADP-ribosyl hydrolases (macroD1, D2 and TARG1) in tumorigenesis. When considering the subject as is, the review is well written, clearly presented and interesting because it summarizes the litterature on the subject. In my feeling, it is just regrettable to restrict the review as done here. The conclusion falls into the finding that "there is limited evidence for a role of macroD1, macroD2 and TARG1 in tumorigenesis" that might limit the impact of the review in the field. Indeed, is it then even timely to review this question ? While I think that the authors have done a good job here regarding the subject as is, I would recommend them to extend the subject to make the review more attractive. Possible suggestion: ADP-ribosyl-hydrolases in genome integrity and cancer ?  Why restricting the review to the 3 macrodomain containing ADP-ribosyl hydrolases only ?

Author Response

We thank the reviewer for critically assessing our work. 

We appreciate the suggestion to extend this work, would however like to explain why we have limited the review. We have opted for this, because no recent reviews focused on MacroD1/MacroD2/TARG in cancer exist.  The ADP-ribosylhydrolases (ARHs) have been reviewed only recently, for example: Bu X, Kato J, Moss Biochem Pharmacol. 2019 Sep;167:44-49. doi: 10.1016/j.bcp.2018.09.028. Emerging roles of ADP-ribosyl-acceptor hydrolases (ARHs) in tumorigenesis and cell death pathways. As the title of our review was known to the editorial office from the start, we assumed this topic would be suitable. Lastly, in the same special issue of Cancers a review focused solely on ARH1 was published, indicating that quite narrowly focused articles are acceptable for this special issue.

We fully agree that negative conclusions may not appear as attractive as positive ones, but they are nevertheless relevant. We have critically reviewed many manuscripts that do hint at a role for these hydrolases in cancer, which we dispute. We think that thereby this review has a twofold function: it serves as a good introduction for researchers interested in these three proteins and contains a warning not to simply believe the body of literature that suggests a role of MacroD1/MacroD2 in cancer.

Reviewer 3 Report

In the review entitled “The controversial roles of mono (ADP-ribosyl) 2 hydrolases MacroD1, MacroD2 and TARG1 in 3 carcinogenesis”, Feijs and colleagues discuss the existing literature on the role of hydrolases in cancer, focusing on the macrodomain-containing hydrolases, after briefly describing ADP-ribosylation, MARylation and mentioning the function of PARPs.

The manuscript is a systematic review that addresses the current knowledge on the topic in a scientifically critical manner. It is suitable for publications, there are just few minor concerns, listed below, that should be addressed by the authors.

Minor:

Line 39 Article or articles?

Line 59 Please check “Interferon-“ 

Line 67/105 please make sure that the Greek letters of “-sheet” and “-helices” (and of the other words on the listed lines ) are included in the text.

Line 69-70 please rephrase

Line 164 please rephrase

Line 177-179 The figure legend should be attached to the Figure/Figure title.

Line 192 please add what type of cells the MCF7 are

Line 218-220 please check the sentence

Line 257 please add a comma after “Conversely”

Line 295 please mention what U2OS cells are

Author Response

We would like to thank the reviewer for positive assessment and critical reading of our manuscript. We have addressed all comments in the revised version. The indicated symbols were present in our submitted version, but then lost after formatting. We have written them out in full and will check with the editorial office that they get replaced again with the symbols before publishing.